# 'It is what was handed over to us as our heritage and must not be taken away just like that': Traditional birth attendants' attitudes towards the elimination of intergenerational female genital mutilation/cutting in Osun State, Nigeria

**Rosemary Omolara Fafowora**⊙*, **Sinegugu Evidence Duma**

Department of Nursing, School of Nursing and Public Health, College of Health Sciences, University of KwaZulu-Natal, Durban, South Africa

* rosemary.olagunju@gmail.com

**Data Availability Statement:** All relevant data are within the paper and its Supporting information file.

## Abstract

Female genital mutilation/cutting is a harmful practice that violates the sexual and reproductive health rights of women and girls. The practice is often perpetrated directly or indirectly from one generation to another as a way of preserving the culture, thus making it difficult to tackle using ordinary prevention interventions. The purpose of the study was to assess the attitude of the traditional birth attendants as community leaders towards the elimination of intergenerational female genital mutilation/cutting (FGM/C) of girls and to determine their level of readiness and preparedness towards achieving it in Osun State, Nigeria. A qualitative research design, using the adapted REPLACE community readiness tool to end female genital mutilation/cutting interview guide, was used to individually interview eight traditional birth attendants who were identified through purposive sampling method as community leaders and key informants. Thematic Analysis was used to analyze the data which yielded female genital mutilation/cutting as traditional heritage, defiance against government efforts and debunking all "western" information about dangers of female genital mutilation to women as lies as findings. The current defensive attitudes of the TBAs as community leaders and custodians of the FGM/C tradition are that of denial and resistance which is characterised by misconception and incorrect knowledge about the issue as well as misconception and lack of support for addressing the issue which is an indication of low level of no readiness for any intervention to prevent or eliminate FGM/C in Osun state, Nigeria. Serious engagement and dialogue between policy makers and health professionals on FGM/C and its effects on women is highly recommended for effective FGM/C elimination strategies to be co-developed with community leaders. Such engagements should adopt a non-confrontational, respectful, and honest approach so as to not alienate the TBAs.

**Funding:** The authors received no specific funding for this work.

**Competing interests:** The authors have declared that no competing interests exist.

## Introduction

Female genital mutilation/cutting (FGM/C) is a public health problem globally. It comprises all procedures that involve partial or total removal of the external female genitalia, or other injury to the female genital organs for non-medical reasons. It is also referred to as multiple procedures on the female genital organs for non-medical reasons [1]. Female genital mutilation/cutting has been described as a harmful practice that violates the sexual and reproductive health rights of women and girls [2, 3]. It is estimated that more than 200 million girls and women alive today have undergone FGM/C across 30 countries including the Middle East, Asia, and Africa [4]. Of this, about 44 million are girls under the age of 15 years [5]. It is further projected that more than 3 million girls and women are currently at risk of FGM/C worldwide [4]. The current estimates of 3.9 million girls mutilated annually will also rise to 4.6 million by 2030 because there is an increase in the population growth in the communities that still practice FGM/C [6]. It is reported that 20 million girls and women are estimated to have undergone FGM/C in Nigeria, with Osun state, where the study was conducted, having reached the highest prevalence of 76.6% of FGM/C [7].

In Nigeria, FGM/C is a deeply entrenched custom that is commonly performed by the traditional birth attendants (TBAs) who claim it as a heritage which confers honor on a woman and her family, a rite of passage to womanhood, preservation of chastity, enhancement of fertility, and prevention of death of the child during delivery among its benefits [8].

Despite its cultural importance, FGM/C has received considerable criticism worldwide because of its potential risk for various forms of health complications [9]. The short term and physical health complication of FGM/C includes but are not limited to pain, risk of hemorrhage, shock, tetanus/sepsis, inability to urinate, damage to adjacent organs such as the rectum and urinary tract, dislocation and fracture of the pelvic bones and the head of the femur due to struggle while being restrained, open sore in the genital region, risk of HIV/AIDS or Hepatitis B. While the long term and reproductive health consequences of FGM/C include difficulty in passing menstrual blood, vaginal fistula, painful sexual intercourse, and difficulty in passing urine which can lead to recurrent urinary tract infection, clitoral cyst, infertility, an increased risk of childbirth complication and newborn death, need for future surgeries to allow for sexual intercourse and childbirth, psychological trauma, and risk of death from hemorrhage or infection [10].

WHO (2023) asserted that FGM/C constitutes a grave violation of the rights and bodily integrity of girls and women because it is predominantly occurring from infancy to the age of 15 when the victims cannot decide for themselves [11]. Furthermore, sexuality is an equal right that both males and females must benefit from. While males are being circumcised for health and religious reasons, women and girls are cut deliberately to remove their sexual feelings and subjugate them to the dictates of their men counterpart, and it is done without any consent from them.

In 2005, the Violence Against Persons (Prohibition) Act 2015 was passed to eliminate the FGM/C practice in Nigeria. Since then, several legal, community based, and health-based strategies had been instituted to eliminate the FGM/C practice [12]. These strategies were met with strong opposition from the TBAs, who remain the self-proclaimed custodians of the traditional responsibility of FGM/C from time immemorial, especially in Osun State, Nigeria. The intergenerational preservation of FGM/C is reflected on most women's belief that the elimination of FGM/C is tantamount to defaulting from their religion or traditions which could breed calamity for them and their offspring. These beliefs have been passed down from one generation to the other and contribute to the high prevalence of FGM/C in Osun State, Nigeria, where it has become major public health issue that requires public health approach-based

prevention interventions. Because of their influence on the intergenerational preservation of FGM/C, we identified the TBAs as community leaders who are key towards the development and implementation of public health prevention intervention for FGM/C in Osun State, Nigeria. The purpose of the study was to assess the attitude of the TBAs as community leaders towards the elimination of intergenerational FGM/C of girls in Osun State, Nigeria, in order to determine their level of readiness and preparedness towards achieving its elimination.

## Materials and methods

### Study setting

Osun State is an inland state in South-West Nigeria with a population of 3,423,535 [13]. The present Osun State was created in 1991 from the old Oyo State. The state shares boundary with Kwara State in the North, Oyo in the West, Ogun and Ondo in the South and Ekiti State on East. The state capital is Osogbo, which is a local government area on its own. In all, a total of thirty local government council areas and Ife east area council constitute the state. Among the major towns in the state are Ilesha, Ife, Ede, Ikirun, Ikire, Ejigbo and Osogbo [14]. Before this time, the state was part of the old Oyo State. Osun State experiences a tropical climate while the local vegetation is the lowland rainforest type.

Indigenes of Osun State belong to the Yoruba race and comprise the Oyo, Ife, Ijesa, Igbomina and Osun. Many non-indigenes and foreigners also reside in the state and live together in harmony. Yoruba and English are the languages used for official and business transactions. The people have a rich cultural heritage that is demonstrated in all spheres of their life including the performance of FGM/C. Their culture finds expression in their arts, literature, music, and other social activities.

This study was conducted among the Yoruba TBAs residing across the three senatorial districts in Osun State, Nigeria. The Yoruba people have a rich cultural heritage that is demonstrated in all spheres of their life including the performance of FGM/C in the Osun State, Nigeria.

### Research design

A qualitative research design adapted REPLACE community readiness to end FGM/C interview guide [15] was used to assess the attitudes of the TBAs as community leaders towards elimination of intergenerational FGM/C to determine their level of readiness and preparedness towards achieving its elimination in Osun State, Nigeria. This was the most suitable approach because it focuses on social and cultural contexts to understand how a problem is perceived and whether taking some type of action is seen as needed by community leaders to change the problem [16, 17]. The adapted REPLACE community readiness to end FGM/C interview guide [15] was translated to the local (Yoruba) language by a bilingual expert in Yoruba and English who is also a qualitative researcher for comparison of the language translation to avoid the loss of meaning in the translated interview guides. The same person was used for translating transcripts to English.

### Sampling and recruitment of participants

Purposive sampling method was used select eight TBAs whom we were regarded as community leaders and key informants with rich information regarding intergenerational FGM/C in Osun State, Nigeria. The inclusion criteria for recruitment and participation in the study included only TBAs who were formally recognized as leaders in performing FGM/C, who had been previously involved in any of the Government's organized seminars or workshops on

FGM/C elimination and or prevention intervention in Osun State, irrespective of the duration of the practice. The selection and recruitment of the participants commenced once the ethical approval from the Humanities and Social Sciences Research Ethics Committee [HSSREC/ 00000924/2019] of University of Kwa-Zulu Natal, South Africa, and Osun state Ministry of Health [OSHREC/PRS/569T/157] was obtained. In Nigeria, integrating the TBAs into the conventional maternal health-care delivery systems, has led to a rise in the utilization of health care facilities because majority of the women still prefer to patronise the TBAs than the skilled health workers though they are trained to subsequently direct their patients to the health centres for appropriate medical care and management [18]. However, the connection between the TBAs and FGM/C is significant as most often than not, they are involved in carrying out the practice in the communities where this harmful practice is widespread. Hence, the reason for choosing the nurses working at the primary health centers withing the research settings were chosen as the gate keepers. Meetings were held with the nurses in the maternity units across the three senatorial districts as gatekeepers to inform them about the purpose of the study and to seek their permission to meet the TBAs working within the location of their primary health centers. A meeting with TBAs was held where the purpose of the study, the research eligibility criteria and request for voluntary participation from those who meet the inclusion criteria were discussed. Those who showed interest were recruited and were given the information sheet and the informed consent form to complete and sign. Arrangements for the interview session were then made by allowing the potential participants to choose suitable date and time within the first 7 days after the participants have given their consent. Before the commencement of the interview, the purpose of the study, and permission to audio record the interview sessions were explained again to the participants in the local language (Yoruba). A total sample size of eight traditional birth attendants as community leaders and key informants were recruited in the study. This small sample size is in line with the REPLACE community readiness tool, which suggests 6–12 key informants as sufficient to address issues in the community [15].

### Pilot study

The first two TBAs out of the sample size were used for pilot study to ensure that the interview guide questions were easy and yielded the necessary data. The data generated from the pilot study were included in the main study data analysis. Contamination of data was not considered to be a concern because in qualitative studies data collection is progressive and the data from pilot studies can be used for the main study [19, 20].

### Data collection

Data collection and data analysis were conducted concurrently between October 2020 and March 2021. All COVID-19 precaution protocols like using the face mask, maintaining social distancing and hand sanitizing were observed during the interviews. In-depth interviews were conducted with the individual TBAs in their homes by the first author in Yoruba, the local language, using the interview guide adapted from the REPLACE Community Readiness to End FGM/C Assessment tool [15]. The interviews were audio recorded and transcribed within 24 hours of recording when information was still fresh in the researcher's memory for data analysis purposes. Each interview lasted between 45–60 minutes to complete thus allowing each TBA to adequately share their views as attitudes towards elimination of intergenerational FGM/C in Osun state. Nigeria. The transcripts were stored in word document formats on a computer with encrypted password and was backed up to an external hard drive to prevent data loss in line with the University's principles for data storage.

## Data analysis

The six steps, namely: familiarization, coding, searching for themes, reviewing themes, defining, and naming themes, and writing up as described by [21] were used to analyze the transcribed data.

The first author went through the transcripts reading and re-reading them to familiarise herself and make sense of the data. Notes were taken and preliminary ideas for codes were marked. In order to start organizing the data into meaningful groups, codes were assigned to briefly describe something interesting that was said by the participants. The first author grouped the newly generated codes with same meaning together to form themes. The second author, who also doubled as the inter coder at this stage, also did the same thing. The generated themes by the first and second authors were scrutinized by both of them to ensure that they were useful and accurate representation of the data by discarding some themes and creating new ones from the transcribed data. The first author then formulated the meaning of each theme so that the data can be understood. At this stage, appropriate naming of themes was made to succinctly interpret the data collected.

## Ethical permission

All ethical principles were adhered to including, consent form for voluntary participation, ethical approval from the Humanities and Social Sciences Research Ethics Committee [HSSREC/00000924/2019] of University of Kwa-Zulu Natal, South Africa, and Osun state Ministry of Health [OSHREC/PRS/569T/157] were obtained.

The names assigned to each of the participants are not their actual names as we are aware of the ethical implication. However, using pseudonyms instead of participants' actual names ensured confidentiality and anonymity.

## Measure to ensure trustworthiness

Measures to ensure trustworthiness and rigor of the findings were applied through the criteria of credibility, confirmability, dependability, and transferability [22].

To ensure credibility, comprehensive field notes, noting gesture and other non-verbal ques observed during the interviews were recorded and analyzed. Audit trail of all research processes were recorded. Analysis of the data gathered reflected a neutral interpretation of the data rather than the researcher's view as recommended by [23, 24]. This was achieved by conducting a member checking on data whereby the analyzed data was summarized and provided to the participants for them to validate and determine the accuracy of the data as their views. For dependability and transferability, an audit trail was kept and the comprehensive and rich details of the study setting, and the socio-demographic characteristics of the participants are presented. To further establish trust and confidence in the findings, the authors compared their individually generated themes from the raw data to ensure that they were useful and accurate representation of the data by discarding some themes and creating new ones from the transcribed data. The senior author and research supervisor reviewed the themes against raw data and helped in refining these to reflect participants' views to enhance confirmability. Common issues recur and main themes were identified to summarize all the views that have been captured in the data. One theme and three subthemes emerged that revealed the defensive attitude of the TBAs towards the elimination of intergenerational FGM/C of girls in Osun state, Nigeria.

**Table 1. Participants' demographic characteristics.**

| Participant | Age | Gender | Religion | Educational level |
|---|---|---|---|---|
| Lukman | 45 | Male | Islam | Tertiary education |
| Abike | 70 | Female | Islam | Primary education |
| Adeyemi | 53 | Male | Islam | Tertiary education |
| Olalekan | 86 | Male | Islam | Primary education |
| Mojirola | 63 | Female | Christian | Secondary education |
| Amope | 80 | Female | Islam | Primary education |
| Fadipe | 73 | Male | Islam | Secondary education |
| Adebayo | 75 | Male | Islam | Primary education |

## Results

Eight (8) TBAs, comprising of three females and five males participated in the study on TBA's attitude towards the elimination of intergenerational FGM/C of girls in Osun State, Nigeria.

These demographics reflects the TBAs in Osun State where the person who becomes a TBA is culturally determined through the lineage of the traditional circumcisers. It is a taboo for a woman who is married to perform FGM/C as TBA, but she can still retain her knives for performing FGM/C on her own children only (personal conversation with Mr Adeyemi, a male participant to the study, December 2020). This explains why most participants were male TBAs in the study as reflected in Table 1. Their ages ranged between 45 and 86 years. Seven of them practiced Islam and only one was Christian as a religion.

### Defensive attitude towards elimination of intergenerational FGM/C

Analyzed data yielded "Defensive attitude towards elimination of FGM/C" as the main theme in the TBAs' attitude towards the elimination of intergenerational FGM/C of girls in Osun State, Nigeria. Three subthemes, including (i) FGM/C as a traditional heritage, (ii) Defiance against government's efforts to eliminate intergenerational FGM/C and Debunking all "western" information about dangers of FGM/C as lies emerged under the main theme.

### FGM/C as traditional heritage

The FGM/C as a traditional heritage sub theme emerged from data related TBAs' defensive attitudes towards elimination of intergenerational FGM/C as traditional and cultural beliefs that the practice is the identity and the only link between the current and past among generations as described in the following extracts.

The response from the respondent below upheld the fact that FGM/C is their traditional heritage that cannot be stopped in their lifetime.

*I am talking about our traditional heritage here. It can't be stopped; it can't stop now. The law, civilization coupled with ignorance is stopping people, but it cannot be stopped.*

[Lukman, Male, Age: 45, Tertiary Education]

Another respondent also mentioned that FGM/C is a traditional heritage that was handed down by their forefathers.

*I think all they are saying about stopping of our tradition is done out of civilization and hatred, hatred of us and our traditional heritage. What has been established by our forefathers cannot perish. It is just like asking us to abandon our language because of civilization. There is no where we meet that we can do without speaking our language. Girl child circumcision cannot perish.*

[Abike, Female, Age:70, Primary Education]

Further, data under this subtheme showed that TBAs considerate as their primary responsibility to prevent intergenerational practice of FGM/C from extinction as depicted in the following extract.

*Girl circumcision has been in existence from time immemorial. Therefore, it can be described as our cultural identity that was transmitted from generation to generation that no one is ready to allow it to die down in his or her own time, even if your government says it is bad for us.*

[Fadipe, Male, Age:73, Secondary Education]

It was also reported as shown in the extract below that the TBAs have a duty to protect FGM/C being their inheritance.

*But you know, it is our heritage which must be promoted. It is not that it really makes more money for us. Do you understand me? It is not what the TBAs are eating from. But it is what was handed over to us as our heritage and must not be taken away just like that!*

[Abike, Female, Age: 70, Primary Education].

## Defiance against government's efforts to eliminate FGM/C

This subtheme emerged from data related to the defiant attitudes of the TBAs towards seminars and workshops that had been previously arranged by government and other organizations in the country with the aim of highlighting the disadvantages of intergenerational FGM/C proposals towards eliminating FGM/C as described in the following extracts.

The extract below shows that despite the attendance at the seminar organized for the TBA by the government, stopping FGM/C is going to be a disadvantage to the women because their sexual urge may not be easily controlled as we are seeing around us.

*I attended the workshops about this, but I was and am still convinced that the disadvantages of government's effort to stopping FGM/C will be enormous for our women. See what is happening nowadays in our communities. No one will be able to control these girls. We must fight anyone including the government to protect our girls from promiscuity.*

[Lukman, Male, Age: 45, Tertiary Education]

The following extract upholds the submission by the TBAs that FGM/C prevents promiscuity which explains why the tradition has been successfully continued.

*No, I don't agree with the government's decision. The result will be very bad. As for me, I did it for all my children both boys and girls. None of my children are promiscuous because I circumcised them and took care of them all. I buried what was cut and properly watered the soil where I buried it so that chicken could not be able to unearth it. Otherwise, if unearthed, the*

*girl child will be promiscuous. We are very careful about these things so that no one can come back and say FGM/C doesn't work.*

[Amope, Female, Age: 80, Primary Education].

A respondent bluntly declared that she will never support the government in stopping FGM/C as depicted in the following extract.

*Ah! I am not in support of the government stopping the girl child circumcision. That is our campaign, and I am not hiding it. At least, it will not happen in my time.*

[Abike, Female, Age 70, Primary Education]

Elimination of FGM/C is described as an advantage to the girl child that no one would want to support as shown in the quote below.

*Personally, I don't agree with government's reasons to stop girl child circumcision. I have said it earlier that, based on my experience and the level of my education, and coming from a family that is traditionally practicing FGM/C, I do not see it being stopped by anyone, even the government or anyone else, not now. I did it (FGM/C) for my girls and there is no problem associated with it, I don't think it should be stopped. It will be a great disadvantage if anyone could agree with government.*

[Lukman, Male, Age: 45, Tertiary Education].

## Debunking all "Western" information about dangers of FGM/C to women as lies

This subtheme emerged from data related to TBAs efforts in debunking most of the information and claims about the benefits of eliminating intergenerational FGM/C as just government's efforts to brainwash people, against the practice of FGM/C as depicted in the quotes below.

The information about the girls having reproductive challenges later in life was debunked by the respondent as shown in the quote below.

*The girls that I circumcised did not have any problem. Even when grown up, these girls do not develop any of the problems we were told resulted from FGM/C, which we were now told can be associated with FGM/C. The girls are now big women and are all alive and doing well in their families. They got married to nice men and have lively children.*

[Amope, Female, Age 80, Primary Education]

The extract below shows that the respondent disagrees with the record of death sequel to FGM/C. He said that the girls he circumcised as infants are still alive today.

*Every girl that we have circumcised is still alive even today. No child can die in the hand of circumcision surgeon and all our girls were circumcised. Anyone who say we put our girls in danger is a liar. My wife is circumcised, all my girls are circumcised and alive. They must stop spreading lies about FGM/C.*

[Olalekan, Male, Age 86, Primary Education]

In addition, the likelihood of lack of enjoyment during sex was further refuted but rather held that circumcised women enjoy sex more as shown in the quote below.

*That debunked FGM/C leads to women not enjoying sex is one of the lies the government is peddling about girl circumcision. But they have forgotten that it's only a small part of the clitoris that is cut and not the whole clitoris. it is not true at all.*

[Adeyemi, Male, Age 53, Tertiary Education]

*It is only a false assumption that circumcised women do not enjoy sex. When a circumcised woman has sexual intercourse with men, she will glue to him as if they tie penis and vagina together, in that way, they both enjoy each other nicely*

[Olalekan, Male Age: 86, Primary Education]

Furthermore, the views that FGM/C have complications was refuted as just rumors as captured by the following extracts.

*There is nothing like transmission of sexual diseases by FGMC, bleeding and all so many other things that are being said about female circumcision. It is safe and cultural to do it (circumcision).*

[Mojirola, Female, Age 63, Secondary Education]

*We were also told that female circumcision causes sexual transmitted diseases / infections which is not true at all! Honestly, I don't know what the government want to gain from spreading such lies for stopping it. Female circumcision does not have any negative side effects as it is being peddled around.*

[Amope, Female, Age 80, Primary Education].

## Discussion

The study was conducted to assess TBAs' attitudes towards the elimination of intergenerational FGM/C of girls in Osun state, Nigeria in order to determine their level of readiness and preparedness towards achieving its elimination. It is important to note that TBAs attitudes reflect a broader societal attitudes and cultural norms that is prevalent in the region.

In recent years, there have been significant efforts to promote the elimination of FGM/C in Nigeria. These efforts include education and awareness campaigns, legal measures, and community engagement programs. However, despite these efforts, our data reveals that the TBAs are still holding on to the traditional beliefs which support the practice.

Our findings show a deep defensive attitude towards elimination of FGM/C among the TBAs as community leaders and custodians of the intergenerational FGM/C which is reflected in the participants views that FGM/C is their traditional heritage and its elimination can be equated to erasing their cultural identity. Similar conclusion was made by Dunn (2018), whose research on how possible it is to end female circumcision in Africa reported that the cultural identity of the Africans was eliminated by current civilization [25]. Sakeah et al (2019) further identified tradition/culture as one of the contributory factors to the persistent practice of FGM/C in Ghana despite it being considered illegal in that country [26]. Our findings revealed the TBAs belief that FGM/C is a cultural tradition that must not be taken away in a similar way that other traditions were eroded by the colonial masters who gradually stripped their subjects off their cultural heritage.

The findings on TBA's strong defiance against the government's prevention interventions such as seminars and workshops organised by the government on the elimination of intergenerational FGM/C is contrary to the findings by Andarge M. Y. (2014) who reported that unsuccessful elimination of intergenerational FGM/C in Ethiopia was due to the government's inability to involve the major community stakeholders such as the TBAs [27].

If not properly handled, the TBAs' defiance against governments' efforts to eliminate FGM/C, can result in a continuous non-alignment and disagreements between government agencies and those responsible for promotion of historic traditions (such as FGM/C) as was reported by Dunn (2018) [25].

According to the TBAs all the government's efforts to eliminate FGM/C was perceived as the government's efforts to brainwash people, thus debunking all information and claims about the benefits of eliminating FGM/C as reported previously by Ibrahim I.A et al. 2013 where it was reported that more patients with complications of FGM/C such as difficult delivery, perineal laceration, hemorrhage, scar/keloid etc. were seen by nurses/midwives than doctors, thus indicating that eliminating FGM/C will be beneficial in preventing these complications [28]. The TBA's lack of knowledge about complications of FMG/C may be reflected as negative attitudes towards elimination of intergenerational FGM/C. On the other hand, holding the viewpoint that the complications from FGM/C are due to the participation of informal "fake" TBAs is a dangerous viewpoint that needs to be corrected if FGM/C is to be eliminated. individuals who are not typically burdened with such responsibilities.

## Strength and limitation

This study used a qualitative approach thus encouraging the TBAs as custodians of FGM/C traditions to voice their views on FGM/C freely with the researchers. Such openness helped in identifying deeply held beliefs that need to be addressed for effective elimination of FGM/C at community level. However, the fact that the gender and profession of the researcher (a woman and a nurse) could be perceived as a limitation because most TBAs were men. Some of their responses could be seen as men exerting their dominance on women.

## Conclusion

The current defensive attitudes of the TBAs as community leaders and custodians of the FGM/C tradition are that of denial resistance which is characterised by misconception and incorrect knowledge about the issue as well as misconception and lack of support for addressing the issue which is an indication of low level of no readiness for any intervention to prevent or eliminate FGM/C in Osun sate, Nigeria. Serious engagement and dialogue between policy makers and health professionals on FGM/C and its effects on women is highly recommended for effective FGM/C elimination strategies to be co-developed with community leaders. Such engagements should adopt a non-confrontational, respectful, and honest approach so as to not alienate the TBAs. Based on the study findings, it is recommended that the TBAs too, in conjunction with the members of the same community, should prove that they did not personally experience or know of girls or women who have undergone FGM/C that developed health complications.

## Supporting information

**S1 File. Transcripts of the interviews of the participants.**
(DOCX)

## Acknowledgments

Our sincere appreciation goes to the management of the Ministry of Health in Osun state Nigeria and University of KwaZulu-Natal, South Africa for granting us the ethical approval to carry out the study. We also thank the Nurses and the Traditional Birth Attendants for creating time to grant the interviews.

## Author Contributions

**Conceptualization:** Rosemary Omolara Fafowora, Sinegugu Evidence Duma.

**Data curation:** Rosemary Omolara Fafowora.

**Formal analysis:** Rosemary Omolara Fafowora, Sinegugu Evidence Duma.

**Methodology:** Rosemary Omolara Fafowora.

**Supervision:** Sinegugu Evidence Duma.

**Validation:** Sinegugu Evidence Duma.

**Writing – original draft:** Rosemary Omolara Fafowora.

**Writing – review & editing:** Rosemary Omolara Fafowora, Sinegugu Evidence Duma.

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
