## [Decision Letter · Decision Letter 0]

26 Sep 2022

PGPH-D-22-01175

'It Is What Was Handed Over to Us as Our Heritage and Must Not Be Taken Away Just Like That': Traditional Birth Attendants' Attitudes Towards the Elimination of Intergenerational Female Genital Mutilation/Cutting in Osun State

Dear Dr. Fafowora,

Thank you for submitting your manuscript to PLOS Global Public Health. After careful consideration, we feel that it has merit but does not fully meet PLOS Global Public Health’s publication criteria as it currently stands. Therefore, we invite you to submit a revised version of the manuscript that addresses the points raised during the review process.

We look forward to receiving your revised manuscript.

Kind regards,

Ditte S Linde, Ph.D.

Academic Editor

Journal Requirements:

Additional Editor Comments (if provided):

Thank you for submitting your manuscript to PLOS Global Public Health. Your paper has now undergone review, and recommend that you make a thorough revison of your manuscript based on the reviewer's comments and resubmit a revised version of manuscript.

Reviewers' comments:

Reviewer's Responses to Questions

**Comments to the Author**

1. Does this manuscript meet PLOS Global Public Health’s publication criteria? Is the manuscript technically sound, and do the data support the conclusions? The manuscript must describe methodologically and ethically rigorous research with conclusions that are appropriately drawn based on the data presented.

Reviewer #1: Yes

Reviewer #2: Partly

2. Has the statistical analysis been performed appropriately and rigorously?

Reviewer #1: Yes

Reviewer #2: Yes

3. Have the authors made all data underlying the findings in their manuscript fully available (please refer to the Data Availability Statement at the start of the manuscript PDF file)?

Reviewer #1: Yes

Reviewer #2: Yes

4. Is the manuscript presented in an intelligible fashion and written in standard English?

Reviewer #1: Yes

Reviewer #2: Yes

5. Review Comments to the Author

Reviewer #1: Thank you for getting the chance to review such sensitive issues! The manuscript has shown the status of the community leaders, called the TBAs aggressive response and stand on FGM. It is really amazing findings how the health system is in danger to protect the population from harmful practices. Though the manuscript couldn’t say any thing about the relation between the TBAs and the country health system, I thought they may have long time relationship and challenging to bring behavioral change on the pre-set mind belief.

The respondent’s openness and strong stand on their belief can help the policy makers and programmers to re-design different interventions and strategies.

Reviewer #2: Thank you for inviting me to review this manuscript and here are some of comments

General comments

There are many editorial issues.

Issues related to chronological arrangements.

Abstract, method and discussion part needs major revision.

ABSTRACT

Fully adhere to the PLOS Global Public Health Submission Guidelines.

There are more than 300 words in the abstract.

The abstract section cannot contain any abbreviations.

The abstract should essentially be divided into three parts: Background, Methodology/Results, and Conclusions/Significance.

The Data Availability Statement, funding, and competing interests should not be included in the abstract section; instead, they should be included after the acknowledgement.

Try to include the severity, gab, and the rationale for the study towards the end of the background.

It doesn't seem like a reasonable objective to "identify and describe attitudes."It appears to be a goal for a quantitative study.

What were the main conclusions of your study, its sampling strategy, and design?

Your findings should guide your recommendation..

Generally the abstract generally does not adhere to the PLOS Global Public Health Submission Guidelines.

INTRODUCTION

Try to start from global to local (what appears to be the burden of FGM- worldwide to Africa, and to the study area) by following the chorological arrangement.

Try to mention the rationale for the study's conduct and the findings of your research in the concluding paragraph.

METHODOLOGY

Study area

Try to provide a reference and some background information on the research area.

Study design

Your study's methodology is unclear. Which qualitative study design types do you employ?

Is it appropriate from a scientific standpoint to use TBA as a key informant and community leader?

Indicate your population and exclusion criteria in clear terms.

Ethics issue

The ethical issue did not cover the option to decline to take part in the study.

The validity of qualitative studies

It is preferable to include the trustworthiness criterion for qualitative research under data quality control. Your study has a problem with credibility. The reason being that data should be acquired from various people in accordance with Guba's Four Criteria for Trustworthiness for Qualitative Research, but you only collect from TBA. The participant has the right to decline to participate in the study, but this was not considered an ethical issue. The study has to present different methods of data collecting, but you only employ one technique of data gathering. In general, method and data triangulation were overlooked.

RESULT

It is unethical to publish a study participant's name.

DISCUSSION

Try to compare your findings to other studies done around the world and provide scientific justification for the similarities or differences. The second paragraph is not required in the discussion section. Generally, it needs revision

RECOMMENDATION

Your recommendations should be based on your findings, and it is better to recommend specific organizations’.

REFERENCE

Some of your references appear to be out-of-date and unpublished work.

Follow Vancouver referencing method

6. PLOS authors have the option to publish the peer review history of their article (what does this mean?). If published, this will include your full peer review and any attached files.

**Do you want your identity to be public for this peer review?** For information about this choice, including consent withdrawal, please see our Privacy Policy.

Reviewer #1: No

Reviewer #2: **Yes: **mickiale hailu tekle

---

## [Editor Report · Decision Letter 1]

6 Dec 2022

PGPH-D-22-01175R1

'It Is What Was Handed Over to Us as Our Heritage and Must Not Be Taken Away Just Like That': Traditional Birth Attendants' Attitudes Towards the Elimination of Intergenerational Female Genital Mutilation/Cutting in Osun State

Dear Dr. Fafowora,

Thank you for submitting your manuscript to PLOS Global Public Health. After careful consideration, we feel that it has merit but does not fully meet PLOS Global Public Health’s publication criteria as it currently stands. Therefore, we invite you to submit a revised version of the manuscript that addresses the points raised during the review process.

We look forward to receiving your revised manuscript.

Kind regards,

Ditte S Linde, Ph.D.

Academic Editor

Journal Requirements:

Additional Editor Comments (if provided):

Dear authors

Thank you very much for revised article. The article has improved greatly, yet we kindly request you to revise article further in order for us to be able to consider it for publication. It is of key important that you attach a proper revision note with your next resubmission, where you make a point-by-point response to each of the comments for your paper and provide a specific answer of how you have addressed each comment in your manuscript.

Please address the following comments:

Major comments:

1. Results: You have many interesting results, yet they are difficult to read as you state 3-4 long quotes after one another. Please provide elaborating text in between quotes that summarises the key point that you want to highlight.

2. Discussion: Please thoroughly revise your discussion. Currently, your discussion is one long summary of your results. Your discussion should include: "Short summary of key findings", "Study strengths and limitations" and "Comparison with literature".

Minor comments

3. You state in your introduction that men are only circumcised for "health" reasons, yet this is only one element. Male circumcision is also done for religious purposes within some religions. Please specify this.

4. Both authors are affiliated with a South African University, yet the study was conducted in Nigeria. Were no Nigerian researchers involved in the study? And if so, why not? Please specify.

5. It appears from the methodology section that back-and-forth translation was not conducted on the interview guide. Is the correct? If so, this should be stated as a limitation in the discussion.

6. Please specify the mean time of the interviews as well as the range of minutes that the interviews took [min-max].

7. Please add author contributions to paper. It is unclear who conducted the interviews and what each of you contributed with to the paper.

8. Please provide a map (that you and this journal are allowed to reference) that shows the provinces of the study. It will make it easier for the reader to understand where the study was conducted.
---

## [Editor Report · Decision Letter 2]

22 May 2023

PGPH-D-22-01175R2

'It Is What Was Handed Over to Us as Our Heritage and Must Not Be Taken Away Just Like That': Traditional Birth Attendants' Attitudes Towards the Elimination of Intergenerational Female Genital Mutilation/Cutting in Osun State

Dear Dr. Fafowora,

Thank you for submitting your manuscript to PLOS Global Public Health. After careful consideration, we feel that it has merit but does not fully meet PLOS Global Public Health’s publication criteria as it currently stands. Therefore, we invite you to submit a revised version of the manuscript that addresses the points raised during the review process.

We look forward to receiving your revised manuscript.

Kind regards,

Ditte S Linde, Ph.D.

Academic Editor

Journal Requirements:

Additional Editor Comments (if provided):

Thank you for the revised manuscript. The paper has merit, however, there are still minor issues, which we request you to address:

1. The discussion is rather weak and needs to be strengthen. You have only revised the discussion very slightly and not thoroughly as suggested in the previous review. Please revise your discussion thoroughly so that it contains the following elements: (1) short summary of key findings (2) strength and limitations of your study (3) discussion of findings according to other literature. Currently, you have only added a small section to your discussion, which states:

"Limitation

Using local language to collect data could have distorted the presentation of the questions but

back and forth translation of the questions minimized the language bias."

Please properly outline key strengths and limitations of your study in your discussion. If in doubt of how to do so, look at other published articles from this journal for inspiration of how to structure your discussion.

2. In the discussion, you several times use a phrase such as "conclusion of [26]". However, it is unclear to the reader which study "reference 26" is. Please revise this throughout the discussion and briefly describe the study you refer to and main author name so that the discussion can be read without having to look up the reference.

3. Thank you for adding a map to the paper, however, please replace it with another map of better quality that outlines the location of the study in Nigeria instead a of map of the exact region.

4. Please thoroughly outline what each author has contributed with under "author contributions" instead of merely repeating that "The two authors have equal contribution to the work". Please describe who designed the study, who collected the data, who analyzed the data, who drafted the manuscript, etc.

Thank you.
---

## [Decision Letter · Decision Letter 3]

10 Sep 2023

PGPH-D-22-01175R3

'It Is What Was Handed Over to Us as Our Heritage and Must Not Be Taken Away Just Like That': Traditional Birth Attendants' Attitudes Towards the Elimination of Intergenerational Female Genital Mutilation/Cutting in Osun State

Dear Dr. Fafowora,

Thank you for submitting your manuscript to PLOS Global Public Health. After careful consideration, we feel that it has merit but does not fully meet PLOS Global Public Health’s publication criteria as it currently stands. Therefore, we invite you to submit a revised version of the manuscript that addresses the points raised during the review process.

Unfortunately, the initial Academic Editor became unavailable to assess the latest revision. As such, we invited two new reviewers to comment. They have raised some minor points that would be useful to clarify in order to strengthen the manuscript and more clearly convey the study setting.

We look forward to receiving your revised manuscript.

Kind regards,

Hanna Landenmark

Staff Editor

Journal Requirements:

2. Please provide separate figure files in .tif or .eps format only and remove any figures embedded in your manuscript file. Please also ensure all files are under our size limit of 10MB.

3. Please ensure that all Figure files have corresponding citations and legends within the manuscript. Currently, Figure in your submission file inventory does not have an in-text citation. If the figure is no longer to be included as part of the submission, please remove it from the file inventory.

Additional Editor Comments (if provided):

Reviewers' comments:

Reviewer's Responses to Questions

**Comments to the Author**

1. If the authors have adequately addressed your comments raised in a previous round of review and you feel that this manuscript is now acceptable for publication, you may indicate that here to bypass the “Comments to the Author” section, enter your conflict of interest statement in the “Confidential to Editor” section, and submit your "Accept" recommendation.

Reviewer #3: All comments have been addressed

Reviewer #4: (No Response)

2. Does this manuscript meet PLOS Global Public Health’s publication criteria? Is the manuscript technically sound, and do the data support the conclusions? The manuscript must describe methodologically and ethically rigorous research with conclusions that are appropriately drawn based on the data presented.

Reviewer #3: Yes

Reviewer #4: Partly

3. Has the statistical analysis been performed appropriately and rigorously?

Reviewer #3: N/A

Reviewer #4: Yes

4. Have the authors made all data underlying the findings in their manuscript fully available (please refer to the Data Availability Statement at the start of the manuscript PDF file)?

Reviewer #3: Yes

Reviewer #4: Yes

5. Is the manuscript presented in an intelligible fashion and written in standard English?

Reviewer #3: Yes

Reviewer #4: Yes

6. Review Comments to the Author

Reviewer #3: I appreciate the author's commitment to challenging deep-rooted culture, i.e., FGM/C which is a severe right violation in Africa. what makes fighting FGM/C in the communities difficult is that there are scholars like Fuambai Ahmadu who is an African-origin American anthropologist. In this regard authors' commitment is admirable. But, it is good if the title is short and precise.

Reviewer #4: Very good paper that I enjoyed reading and learning. Congratulations to authors.

I would like to share some comments/feedback to authors.

1. MAJOR COMMENT: The replace readiness framework ( nine stages) or how it was adaptated is not described and methods do not elaborate how the themes identified fed into replace the framework. All of this is missing and hard to see how the study findings relate to replace community readiness stages...

2. Definition of TBA needed to understand whether they are part of health system and work under supervision of nurses. It is not clear why nurses were involved and reader may not be familiar with health system structure in Nigeria.

3. Introduction section should not include opinions or results from study line 113-115 e.g. "he TBAs believe in the supremacy of FGM/C as the traditional heritage, which is concerned about the wellbeing of the people, that it predates modern medicine and therefore should not be stopped. This cultural belief has facilitated the intergenerational preservation of FGM/C and explains their attitude towards the elimination of its practice."

If this was obtained from another study, good to insert reference. Similar comment on sexuality and equal rights, good to enter a reference

4. Sampling and recruitment - who determined their status as "community leader" can some clarity be made on how that was defined. How many nurses were met, how many TBAs were met, how many showed interest..it would be nice to mention these details

5. Positionality of data collectors/investigators not addressed or commented on how it may have affected results

6. Result section: not clear how the conclusion of TBAs is culturally determined and related to lineage (lines 281 - 282) from table 1 demographic data that did not capture anything on lineage

7. Discussion section last paragraph is a sweeping statement to generalize data from 8 individuals to explain that it represents all TBA's in the state and explains the rise of FGM. There is no study findings related to COVID and not sure why this point was raised. Suggest to be more cautious and recognize that this is a sample of potentially similar minded TBAs but this would need to be investigated.

8. Conclusion: Suggest to add recommendation based on study findings is .... to involve people from same community to have dialogue with TBA to bypass “western”agenda and “show” them the health complications that they did not personally experience or do not know of girls/women who have undergone FGM

7. PLOS authors have the option to publish the peer review history of their article (what does this mean?). If published, this will include your full peer review and any attached files.

**Do you want your identity to be public for this peer review?** For information about this choice, including consent withdrawal, please see our Privacy Policy.

Reviewer #3: **Yes: **Estifanos Balew Liyew

Reviewer #4: No

---

## [Decision Letter · Decision Letter 4]

19 Apr 2024

'It Is What Was Handed Over to Us as Our Heritage and Must Not Be Taken Away Just Like That': Traditional Birth Attendants' Attitudes Towards the Elimination of Intergenerational Female Genital Mutilation/Cutting in Osun State

PGPH-D-22-01175R4

Dear Mrs Fafowora,

We are pleased to inform you that your manuscript ''It Is What Was Handed Over to Us as Our Heritage and Must Not Be Taken Away Just Like That': Traditional Birth Attendants' Attitudes Towards the Elimination of Intergenerational Female Genital Mutilation/Cutting in Osun State' has been provisionally accepted for publication in PLOS Global Public Health.

In addition, we noted that it looks like approval was obtained from the Uni of Kwazulu Natal but this is not mentioned in the manuscript, just in the acknowledgements. You have listed two approval numbers, so it's possible that the first one where the affiliation is unnamed is this one - please do clarify this in the paper itself as you complete the final formatting changes, and thank you in advance.

Best regards,

Julia Robinson

Executive Editor

Reviewer Comments (if any, and for reference):

Reviewer's Responses to Questions

**Comments to the Author**

1. If the authors have adequately addressed your comments raised in a previous round of review and you feel that this manuscript is now acceptable for publication, you may indicate that here to bypass the “Comments to the Author” section, enter your conflict of interest statement in the “Confidential to Editor” section, and submit your "Accept" recommendation.

Reviewer #3: All comments have been addressed

Reviewer #5: All comments have been addressed

2. Does this manuscript meet PLOS Global Public Health’s publication criteria? Is the manuscript technically sound, and do the data support the conclusions? The manuscript must describe methodologically and ethically rigorous research with conclusions that are appropriately drawn based on the data presented.

Reviewer #3: Yes

Reviewer #5: Partly

3. Has the statistical analysis been performed appropriately and rigorously?

Reviewer #3: N/A

Reviewer #5: Yes

4. Have the authors made all data underlying the findings in their manuscript fully available (please refer to the Data Availability Statement at the start of the manuscript PDF file)?

Reviewer #3: Yes

Reviewer #5: Yes

5. Is the manuscript presented in an intelligible fashion and written in standard English?

Reviewer #3: Yes

Reviewer #5: No

6. Review Comments to the Author

Reviewer #3: I wish the paper to be accessible to the people while published.

Reviewer #5: We appreciated the authors for trying to address this important punlic health issue through qualitative research .However ,

A. The typebof the qualitative studybis not described.

B. The samples and how quality of data is ensured is not well addressed .

C.The paper in general didn't used the guidelines of the journal.

D.The papaer lacks clarity, conciseness and logical flow.

E. The sections didn't contain what it intends to contain. E.g. Methods/results

F. Use of qualitative software is not stated.

Regards,

7. PLOS authors have the option to publish the peer review history of their article (what does this mean?). If published, this will include your full peer review and any attached files.

**Do you want your identity to be public for this peer review?** For information about this choice, including consent withdrawal, please see our Privacy Policy.

Reviewer #3: **Yes: **Estifanos Balew Liyew

Reviewer #5: No
